Shoe configuration effects on equine forelimb gait kinetics at a walk

Aoun Rita 1 2
Ogunmola Zaneta 1 2
Musso Anaïs 1
Taguchi Takashi 1 2
Takawira Catherine 1 2
Lopez Mandi J. mlopez@lsu.edu 1 2
1 Department of Veterinary Clinical Sciences, School of Veterinary Medicine, Louisiana State University and Agricultural and Mechanical College , Baton Rouge , LA , United States of America
2 Laboratory for Equine and Comparative Orthopedic Research, Department of Veterinary Clinical Sciences, School of Veterinary Medicine, Louisiana State University and Agricultural and Mechanical College , Baton Rouge , LA , United States of America
Parkes Rebecca
Electronic publication date: 2025 Feb 26
Publication date: 2025
Volume: 13
Electronic Location ID: e18940
Received 2024 Aug 27; Accepted 2025 Jan 16
Copyright: ©2025 Aoun et al.
Copyright year: 2025
Copyright holder: Aoun et al.
License: This is an open access article distributed under the terms of the Creative Commons Attribution License, which permits unrestricted use, distribution, reproduction and adaptation in any medium and for any purpose provided that it is properly attributed. For attribution, the original author(s), title, publication source (PeerJ) and either DOI or URL of the article must be cited.
License URL: https://creativecommons.org/licenses/by/4.0/

Keywords: Hoof, Horse, Ground reaction force, Wooden clog, Animal biomechanics, Braking, Propulsion, Bar shoes, Forelimb, Force platform

Funding: Tynewald Foundation, a Charles V. Cusimano Equine Health and Sports Performance Grant Louisiana State University School Department of Veterinary Clinical Sciences This work was supported by the Tynewald Foundation, a Charles V. Cusimano Equine Health and Sports Performance Grant, and the Louisiana State University School Department of Veterinary Clinical Sciences. The funders had no role in study design, data collection and analysis, decision to publish, or preparation of the manuscript.

==============================
The shift in vertical forces on the equine hoof surface by heart-bar, egg-bar, and wooden clog shoes can significantly impact gait kinetics. Hypotheses tested in this study were that vertical, braking, and propulsion peak force (PF) and impulse (IMP) are different while shod with heart-bar, egg-bar, open-heel, and wooden clog shoes, or while unshod, and the resultant ground reaction force vector (GRFYZ) has the longest duration of cranial angulation with open-heel shoes followed by unshod, then egg-bar and heart-bar shoes, and the shortest with wooden clog shoes. Forelimb GRFs were recorded as six non-lame, light-breed horses walked across a force platform (four trials/side) while unshod or with egg-bar, heart-bar, open-heel, or wooden clog shoes. Outcomes included vertical, braking, and propulsive peak forces (PFV, PFB, PFP) and impulses (IMPV, IMPB, IMPP), percent stance time to each PF, braking to vertical PF ratio (PFB/PFV), walking speed (m s−1), total stance time (ST) and percent of stance in braking and propulsion. The magnitude and direction of the resultant GRFYZ vectors were quantified at 5% stance increments. Kinetic measures were compared among shoeing conditions with a mixed effects model (p-value < 0.05). A random forest classifier algorithm was used to predict shoeing condition from kinetic outcome measures. All results are reported as mean ± SEM. Trial speed, 1.51 ± 0.02 m s−1, was not different among shoeing conditions. The PFV was lower with wooden clog (6.13 ± 0.1 N kg−1) versus egg-bar (6.35 ± 0.1 N kg−1) shoes or unshod (6.32 ± 0.1 N kg−1); the PFP was higher with wooden clog (0.81 ± 0.03 N kg−1) versus open-heel (0.71 ± 0.03 N kg−1) or egg-bar (0.75 ± 0.03 N kg−1) shoes or unshod (0.74 ± 0.03 N kg−1), and lower with open-heel compared to heart-bar shoes (0.77 ± 0.03 N kg−1). Both IMP B and IMPV were higher with open-heel shoes (−0.19 ± 0.008 N s kg−1, 3.28 ± 0.09 N s kg−1) versus unshod (−0.17 ± 0.008 N s kg−1, 3.16 ± 0.09 N s kg−1), and IMPV was higher with wooden clog shoes (3.26 ± 0.09 N s kg−1) versus unshod. With wooden clog shoes, PFB/PFV (0.12 ± 0.004) was higher than unshod (0.11 ± 0.004). Percent time to peak PFV, PFB, and PFP, and percent braking time were highest and percent propulsion time lowest with wooden clog shoes. The magnitude of the GRFYZ vector with the wooden clog shoe was the highest among shoeing conditions during the first stance half, lowest during the second stance half, highest during late propulsion, and had the most gradual braking to propulsion transition. Vectors were angled cranially with wooden clog shoes slightly longer than the others. Wooden clog shoes was the only shoeing condition accurately predicted from kinetic measures. Distinct, predictable changes in gait kinetics with wooden clog shoes may reduce stresses on hoof structures. Study results enhance knowledge about shoe effects on equine gait kinetics and cutting-edge measures to quantify them.

Introduction

Horseshoes have been employed since the Roman era when “hippo-sandals” were used primarily for protection against rough and abrasive surfaces (Back & Pille, 2013; Karle et al., 2010; Lawrence, 1898). Shoe shape, composition, and affixation techniques have advanced with customization for protection, performance, and therapy. Mechanisms to alleviate limb pain and stabilize and shield hooves compromised by injury or disease continue to evolve with technology, metallurgy, and experience (Back & Pille, 2013; Butler Jr, 1985; Clarke et al., 2021; Goetz & Comstock, 1985; Karle et al., 2010). Commonly, shoes are designed to alter force distribution over the solar hoof surface, often for targeted stress reduction (O’Grady, 2017; Parks, 2012; Van Heel et al., 2005). However, changes in hoof force distribution during a step cycle can also significantly impact gait kinetics (Eliashar et al., 2002; Roepstorff, Johnston & Drevemo, 1999).

Among commercially available horseshoes, heart-bar, egg-bar, and open-heel tend to be highly represented (Aoun, Takawira & Lopez, 2024). Heart-bar, egg-bar and wooden clog shoes are specifically shaped to shift the ground reaction forces in the palmar/plantar direction, and they facilitate breakover (Hüppler et al., 2016; Rogers & Back, 2003; Rogers & Back, 2007). The continuous shape of both heart-bar and egg-bar shoes increases their surface area compared to a standard shoe, though the heart-bar shoe has greater hoof surface contact due to an extension over the frog. Both shoes are used for a number of functional and therapeutic purposes including alleviating heel pain and support of laminitic hooves (O’Grady & Parks, 2008; Rogers & Back, 2003). The wooden clog shoe is designed specifically for hooves affected by laminitis. It consists of a 1.9 cm-thick hoof-shaped, wooden platform with a perimeter beveled at an angle of 45° from proximal to distal, and the flat ground surface is covered with flexible rubber material for traction (O’Grady & Steward, 2009). A craniocaudal rocking motion from impact through lift off is intended to diminish the torque on the dorsal hoof lamellae by reducing the moment of the deep digital flexor tendon around the distal interphalangeal joint and alleviating breakover stresses (O’Grady, Steward & Parks, 2007; Steward, 2003).

Based on current knowledge, shoe composition and shape both influence equine gait kinetics, and quantification of their impact on individual ground reaction forces (GRF) provides a comprehensive assessment of the effects (Eliashar et al., 2002; Hagen et al., 2017; Pardoe et al., 2001; Roepstorff, Johnston & Drevemo, 1999). The force platform is a validated tool for quantification of GRFs in non-lame and lame horses (Barrey, 1999; Ishihara et al., 2009). Established measures include vertical (V), braking (B), and propulsion (P) peak force (PFV, PFB, PFP) and impulse (IMPV, IMPB, IMPP). The contact time with the ground surface during the step cycle is the total stance time (ST), which can be divided into an initial braking followed by a propulsion phase. The resultant GRF force vector in the Y (craniocaudal)-Z (vertical) plane, the vector sum of horizontal and vertical vectors, represents the primary direction and magnitude of the forces between the hoof and ground surface in the sagittal plane (Clayton & Hobbs, 2019). The dynamic ratio of GRFY/GRFZ as a representation of the coefficient of friction is often measured during the sliding phase of the stance cycle (Pardoe et al., 2001). These kinetic measures make it possible to quantify shoe effects on GRF, temporal step cycle components, and resultant GRFYZ vector angles that, together, compose a complete kinetic gait evaluation.

In addition to shoeing condition, the gait parameters described above are influenced by gait velocity and ground surface characteristics. An increase in speed results in higher PFV, lower IMP, and shorter ST in fore- and hind limbs, and forelimb vertical force curves can have a double-peak at high speeds (Biknevicius, Mullineaux & Clayton, 2004; Dutto et al., 2004; Khumsap et al., 2002; McLaughlin et al., 1996; Weishaupt et al., 2010). The loading rates, vertical and horizontal components of PF, pressure distribution, and phalangeal alignment are also significantly affected by the type and roughness of the ground surface at a trot or a walk (Gustås, Johnston & Drevemo, 2006; Hüppler et al., 2016; Setterbo et al., 2009).

As mentioned above, shoe configuration impacts gait kinetics. Shoeing alone is reported to increase breakover duration (Hagen et al., 2021). A recent kinetic investigation confirmed that the size and shape of caulks to improve traction of iron open-heel shoes differently affect fore- and hind limb equine trotting gait kinetics (Wang et al., 2021). Among assessed measures, forelimb PFV increased the most from unshod with composite plastic-steel shoes, and hind limb PFB increased the most from unshod with composite plastic-steel shoes followed by open-heel shoes with a thin layer of tungsten carbide, then open heel shoes with low profile-high surface area calks (Wang et al., 2021). Differences in pressure distribution on the hoof among shoe shapes have been confirmed in separate investigations (Hagen et al., 2017; Hagen et al., 2016; Hüppler et al., 2016); in one study, pressure peaks occurred at the heels with bar shoes, beneath the ends of the shoe branches at the toe with open toe shoes, and beneath the toe and thin branches of a wide toe shoe in walking horses (Hüppler et al., 2016). A recent in situ study confirmed distinct hoof capsule deformation behavior among open heel egg-bar, and heart-bar shoes in unaffected and laminitic hooves at cyclic loads up to 5.5 × 103 N (Aoun et al., 2023). The novel shape and function of the wooden clog shoe is expected to have distinctive effects on gait kinetics.

To date, quantifiable information about the immediate effects of shoe design on the external forces experienced by the hoof during the stance phase of a walk is limited. Therefore, the primary objective of this in vivo study was to quantify equine forelimb walking kinetics while unshod and with open-heel, egg-bar, heart-bar, or wooden clog shoes. Based on distinct hoof deformation characteristics among shoe configurations during dynamic vertical loading in vitro and unique pressure distribution among shoe shapes in vivo, it was anticipated that equine forelimb kinetic measures at a walk would differ among shoes and while unshod due to dissimilar force distribution on the hoof surface throughout the step cycle (Aoun et al., 2023; Aoun, Takawira & Lopez, 2024; Hagen et al., 2017; Hagen et al., 2016; Hüppler et al., 2016). The first tested hypothesis was that vertical, braking, and propulsion PF and IMP are different while shod with heart-bar, egg-bar, open-heel, and wooden clog shoes, or while unshod. The second hypothesis was based on information that shoeing increases breakover duration, egg-bar, heart-bar, and wooden clog shoes facilitate breakover relative to open heel and unshod, and the hoof contact surface area among the three is least with egg-bar and greatest with wooden clog shoes (Hagen et al., 2021; Rogers & Back, 2007). It was hypothesized that the resultant GRFYZ vector has the longest duration of cranial angulation with open-heel shoes followed by unshod, then egg-bar and heart-bar shoes, and the shortest with wooden clog shoes. The results of this investigation include unique information about the effect of common shoe configurations on walking gait kinetics of non-lame horses to expand the science of horseshoes, and they include quantifiable data on the potential protective capacity of wooden clog shoes for damaged or weak hooves.

Materials and Methods

Ethics and regulations

The study was approved based on the ethical and welfare regulations of the Louisiana State University Institutional Animal Care and Use Committee (IACUCAM-21-157).

Animals and housing

Six horses were selected from the university research herd based on the following inclusion criteria: (1) gelding (n = 4) or mare (n = 2); (2) light breed; (3) 5 to 25 years (15.3 ± 1.5 years, mean ± standard error of the mean (SEM)); (4) 400 to 600 kg (545.6 ± 69.4 kg); and (5) no subjective lameness. A licensed veterinarian performed physical and lameness examinations to rule out illness, trauma, or lameness prior to study enrollment. The lameness evaluation was performed according to the criteria of the American Association of Equine Practitioners lameness scale. Horses were housed in individual stalls (3.6 × 3.6 m) with wood shaving bedding for at least two days prior to and for the duration of the gait trials (5 days/horse). They had free access to water in a bucket (16 L), about 2.7 kg of commercial feed (Strategy, Purina Animal Nutrition LLC, Shoreview, MN, USA), and four kg of Bermuda grass hay, which were all replenished twice daily.

Shoeing

The hooves of the horses were trimmed on a regular schedule, every 6–8 weeks, by a certified farrier as part of standard husbandry protocol for the university herd. All four hooves of horses included in the study were trimmed by a certified farrier about 12 days before the initial force platform evaluation that was performed while horses were unshod. For purposes of this study, shoes were only applied to forehooves. Commercially available open-heel (Fig. 1A, 300.5–350.1 g, Kerckhaert, Vogelwaarde, The Netherlands), egg-bar (Fig. 1B, 342.5–410.9 g, Ironworks, Texas Farrier Supply, Weatherford, TX, USA), heart-bar (Fig. 1C, 342.9–456.9 g, Ironworks, Texas Farrier Supply, Weatherford, TX, USA), and wooden clog (Figs. 1D–1F, 144–181.7 g, Equicast, Inc., Lawrence, KS, USA) shoes were used in the study. As indicated above, the shoe weight varied slightly with size and configuration. Shoes were applied in a random order based on a computer-based random number generator (Microsoft Excel, Microsoft, Redmond, WA, USA). Metal shoes were fastened with two nails per side and nail holes were reused between shoes to avoid multiple holes in the hoof wall. The nails were clinched after placement to seat them in the shoe and bend the surface end toward the hoof. For fixation of the wooden clog shoe, sizes were first determined by the farrier based on the hoof width and length. The frog sulci were packed with a two-part silicon-based impression material (Fig. 1D). Then, with the horse standing on the shoe, two wood screws, preplaced at the medial and lateral quarters, were tightened to hold the shoe in place (EDSS, Equine Digit Support System, Inc., Penrose, CO, USA). Subsequently, aliquots (∼5 ml) of fast setting urethane adhesive (HOOFTITE™, Chem Select, Inc., Rancho Santa Margarita, CA, USA) were placed about two cm apart and one cm from the hoof solar margin around the circumference of the hoof (Fig. 1E). Next, moistened 3″ fiberglass casting material (EquiCast®, Equicast, Inc., Lawrence, KS, USA) was wrapped about the outer circumference of the shoe and hoof, and linear low-density polyethylene was applied on top of the cast to facilitate the curing process (Fig. 1F). It was removed after about 5 min.

Figure 1 Commercially available shoes used in the study.

Force platform data were collected while horses were unshod and then after (A) open-heel, (B) egg-bar, (C) heart-bar, or (D) wooden clog shoes were applied to the forelimbs by a certified farrier in random order. Horses were shod with each shoe type 22–24 h prior to kinetic gait collection. Iron shoes were affixed with two nails per side using the same nail holes for all shoes. A two-part, silicon-based impression material was used to fill the frog sulci (D; purple arrow) prior to application of wooden shoes which were stabilized by compressing hoof tissue under the heads of two screws on each side when the screws were advanced into the wood base (D and E; gray arrows) on each side of the wall. Fast setting resin (E; green arrow) was applied to the hoof wall before fiberglass cast material was added to enclose the dorsal hoof wall and periphery of the wood shoe (F; orange arrow). Linear low-density polyethylene was applied to facilitate curing of the casting material (F; white arrowhead).

Kinetic data collection

Horses were walked by experienced handlers over a 900 × 900 mm force platform (Model BP900900, Advanced Mechanical Technology, Inc., Watertown, MA, USA). The force platform was embedded in a ∼40-meter-long concrete runway for GRF data collection. It had a magnetic cover with a thin layer of concrete on the top surface that was the same color and texture as the runway. Light emitting diode markers (LED, 2 × 1 cm) of an active motion detection system (Codamotion®, Charnwood Dynamics, Ltd., Rothley, UK) were attached to the hoof wall using commercially available hook and loop fasteners (1 × 1 cm, Velcro®, Velcro USA, Inc., Manchester, NH, USA) adhered to the markers and hoof surfaces with cyanoacrylate glue. They were attached at the coronary band, the midpoint, and solar margin of the toe and at the coronary band and solar margin of the lateral and medial quarters and heels (Fig. 2, ODIN V2.0, Codamotion®, Codamotion Ltd). An additional marker was attached to a neoprene support boot (Dura-Tech®, Schneiders, Chagrin Falls, OH, USA) at the lateral surface of the fetlock. Two optoelectronic sensor units (Codamotion®) were placed facing each other and the force platform ∼3 m from the center on the x axis. A fixed origin of a virtual cartesian system coordinate axis was established at the center of the force platform. Data from a total of 240 individual trials was recorded at a sampling frequency of 200 Hz with a commercially available software program (ODIN V2.0, Codamotion®, Codamotion Ltd, Rothley, UK), and data was reduced with custom code (Python, The Python Software Foundation, Beaverton, OR, USA). A trial was considered successful if the forelimb hoof fully contacted the platform followed by the ipsilateral hind limb hoof at a speed between 1 to 2.3 m s−1, a range that included a comfortable walking rate for all horses in the study. The acceptable individual variance in individual horse walking speed among all shoe condition trials was a maximum of ±5%. Trials were rejected if both forelimb and hind limb hooves did not contact the platform, if either hoof was not entirely on the force platform, or if either was within five cm of the perimeter. For each shoeing condition, a total of four successful trials per side (left and right, 8 total) were recorded. Speed was calculated for each trial as the distance between the position of the proximal lateral quarter marker at full stance (resultant GRFYZ vector at 90°) for two consecutive steps (a full stride cycle) divided by the elapsed time between the timestamps for each position as the horse walked across the force platform.

Figure 2 Y–Z resultant force vector data collection.

The resultant force vector in the Y (craniocaudal)-Z (vertical) plane, GRFYZ, magnitude and direction were quantified at 5% increments of the step cycle during each gait trial from heel down (A), through full stance (B), to toe off (C).

Kinetic data reduction

Forces

Vertical, braking, and propulsion PF (PFV, PFB, PFP) were the maximum value of each force curve normalized to individual animal weight (N kg−1). The PFV was the highest peak in the vertical force curve, which was the second peak in this study. The area under the force versus time curve for the duration of the stance, IMP, was also normalized to weight (N s kg−1). Impulse was calculated as: IMPX= ∫0100%FX.dt; with X representing the force direction, vertical, caudal (braking), or cranial (propulsion) (IMPV, IMPB, IMPP), respectively.

Peak braking to vertical force ratio

As a representation of traction, the braking to vertical PF ratio was calculated as PFBPFV; where PFB is the peak braking force and PFV is the peak vertical force (Burwell & Rabinowicz, 1953; Wang et al., 2021).

Fore- to hind limb vertical peak force ratio

The ratio of the sum of PFV between fore- and hind limbs for each shoeing condition was determined using PFV values from combined trials for individual horses. It was calculated as SUMPFVforelimbsSUMPFVforelimbs+PFVhindlimbs∗100 or SUMPFVhindlimbsSUMPFVforelimbs+PFVhindlimbs∗100 (Wang et al., 2021).

Y-Z resultant force vectors

The mean GRFY and GRFZ of all trials for each shoeing condition were used to calculate the resultant ground reaction force vector (GRFYZ) in the Y (craniocaudal)-Z (vertical) plane depicting the magnitude (GRFYZ) and direction at 5% increments of the step cycle from heel impact through full support (PFV) to toe off (Fig. 2, ODIN V2.0, Codamotion®, Codamotion Ltd). Vector magnitude and direction were quantified as previously described, except the precise tail of the vectors (center of pressure on the force platform) was not determined in this report (Hobbs, Robinson & Clayton, 2018; Kambhampati, 2007). The magnitude of the resultant GRFYZ vector in 2D was calculated using the formula: GRFYZ=GRFY2+GRFZ2 where GRFY and GRFZ are the vector components in the craniocaudal and vertical directions, respectively. The direction of the GRFYZ vector relative to the horizontal axis at each increment was determined with the Y–Z coordinates of the vector as cos−1GRFYGRFZ. A vector angle of 0∘ ≤ GRFYZ angle < 90∘  represents a cranial force direction (positive GRFY), GRFYZ angle = 90∘ represents a vertical vector (null GRFY), and 90∘ < GRFYZ angle ≤ 180∘ represents a caudal force direction (negative GRFY). Graphic representations of the data were generated with commercially available software (Microsoft Excel, Microsoft, Redmond, WA, USA).

Temporal parameters

The ST in seconds (s) was the time between initial contact of the hoof with the force platform to lift off based on a minimum threshold of PFV = 50 N. Time to braking, vertical, and propulsion PF (time to PFB, time to PFV, time to PFP) were the times from initial hoof contact to the maximum value (peak) of each force component. Braking time was the time from initial hoof contact to the point where braking transitioned to propulsion (GRFY = 0); propulsion time was from the transition point until hoof-lift off. The time to each PF and braking and propulsion times were each determined as a percent of the ST.

Statistical analysis

Data is presented as mean ± standard error of the mean (SEM). All statistical analyses were performed with JMP Pro version 17.0.0 (JMP Statistical Discovery LLC, Cary, NC, 2022–2023). Outcome measures compared among shoeing conditions included PFV, PFB, PFP, IMPV, IMPB, IMPP, PFB/PFV, ratio of the sum of PFV between fore- and hind limbs, ST, percent time to PFB, PFV, and PFP, and percent braking and propulsion times. Differences among shoes were evaluated using mixed-effects models with horse identification and trial number as random effects and shoeing condition, speed, and their interaction as fixed effects. Interaction between shoeing and speed was evaluated with all mixed models, and none were significantly affected, so the interaction was removed as a covariate. Including speed as a covariate, without its interaction with shoeing, improved the prediction fit for all kinetic variables by a smaller Akaike information criterion (AIC). So, speed was included as a fixed effect in the mixed-effects models used to evaluate differences among shoeing conditions. When the shoeing condition effect was significant, post hoc Tukey’s tests were employed for multiple comparisons across shoeing conditions. Speed among shoeing conditions was evaluated via one-way ANOVA. The normality of the residuals from the parametric models was examined and confirmed by quantile plots. Differences in resultant ground reaction force vector (GRFYZ) magnitude among shoeing conditions were compared at each stance increment with a two-way ANOVA with post hoc Tukey’s tests when the shoe effect was significant for a given increment. A random forest (RF) classifier algorithm was used to predict shoeing condition from kinetic outcome measures. Shoeing condition was the target variable with all kinetic measures as predictive features. Significance was consistently set at p-value < 0.05.

Results

Gait speed

The speed for trials included in the data analysis was 1.51 ± 0.02 m s−1. Specific to shoeing condition, the speed was 1.55 ± 0.03 m s−1, 1.55 ± 0.04 m s−1, 1.42 ± 0.03 m s−1, 1.51 ± 0.04 m s−1, and 1.51 ± 0.03 m s−1 for egg-bar, heart-bar, open-heel, and wooden clog shoes, and while unshod, respectively (Fig. S1). Speed was not significantly different among shoeing conditions.

Ground reaction force components and impulses

The PFV was significantly affected by shoeing condition (p-value = 0.0002, Fig. 3); it was highest with egg-bar (6.35 N kg−1 ± 0.1) shoes followed, in order, by unshod (6.32 N kg−1 ± 0.1), and then with heart-bar (6.27 N kg−1 ± 0.1), open-heel (6.23 N kg−1 ± 0.1), and wooden clog (6.13 N kg−1 ± 0.1) shoes. The PFV with wooden clog shoes was lower than with egg-bar shoes (p-value = 0.0002) or when unshod (p-value = 0.002). Similarly, the PFP was significantly affected by shoeing condition (p-value < 0.0001, Fig. 3). The PFP was higher with wooden clog (0.81 N kg−1 ± 0.03) versus open-heel (0.71 N kg−1 ± 0.03, p-value < 0.0001) or egg-bar (0.75 N kg−1 ± 0.03, p-value = 0.01) shoes or while unshod (0.74 N kg−1 ± 0.03, p-value = 0.004), and lower with open-heel compared to heart-bar shoes (0.77 N kg−1 ± 0.03, p-value = 0.02). The IMPV, significantly affected by shoeing condition (p-value = 0.011), was lower while unshod (3.16 N s kg−1 ± 0.09) compared to with open-heel (3.28 N s kg−1 ± 0.09, p-value = 0.009) or wooden clog (3.26 N s kg−1 ± 0.09, p-value = 0.04) shoes. The IMPB, significantly affected by shoeing condition (p-value = 0.0403), was higher with open-heel (−0.19 N s kg−1 ± 0.008) shoes compared to unshod (−0.17 N s kg−1 ± 0.008, p-value = 0.02).

Figure 3 Peak ground reaction forces (PF) and impulses (IMP) across distinct shoeing conditions.

Braking (A-PFB, D-IMPB), vertical (B-PFV, E-IMPV), and propulsion (C-PFP, F-IMPP) peak forces (A–C) and impulses (D–F) normalized to body weight from horses (n = 6) shod with egg-bar, heart-bar, open-heel, or wooden clog shoes and while unshod. Points in the graph represent individual trials, and the horizontal line in the middle of each set indicates the mean; the numeric mean value along with the standard error of the mean (in parentheses) are shown to the right of each data set. Significant differences between data sets with a line between them are indicated by a symbol beneath the line.

Peak braking to vertical force ratio and vertical peak force distribution

The ratio of PFB/PFV, significantly affected by shoeing condition (p-value = 0.024), was lowest when horses were unshod (0.11 ± 0.004), though only significantly lower compared to when shod with wooden clog (0.12 ± 0.004, p-value = 0.015, Fig. 4) shoes. The forelimb:hind limb ratio of the sum of total PFV was about 60%:40% for all shoeing conditions, and it was not significantly different among them (Fig. 5).

Figure 4 Braking to vertical peak force ratio among shoeing conditions.

The PFB/PFV ratio from horses (n = 6) shod with egg-bar, heart-bar, open-heel, or wooden clog shoes and while unshod. Points in the graph represent individual trials, and the horizontal line in the middle of each set indicates the mean; the numeric mean value along with the standard error of the mean (in parentheses) are shown to the right of each data set. Significant differences between data sets with a line between them are indicated by a symbol beneath the line.

Figure 5 Forelimb (cyan) and hind limb (brown) ratio of the sum of total PFV among shoeing conditions.

Percent forelimb and hind limb vertical peak force distribution from horses (n = 6) shod with egg-bar, heart-bar, open-heel, or wooden clog shoes and when unshod. Mean and standard error of the mean (in parentheses) values are shown above each column.

Temporal parameters

Shoeing condition significantly affected the percent times to PFV, PFB and PFP (p-value < 0.0001); all three were greater with wooden clog shoes versus other shoes and when unshod (Fig. 6). Percent time to PFPwas also greater with open-heel (75.22 ± 1.1%) versus egg-bar shoes (73.65 ± 1.1%, p-value = 0.03). The percent of stance time for braking was greater and that for propulsion was lower in horses with wooden clog shoes than with all other shoes or when unshod (Fig. 7; p-value < 0.0001).

Figure 6 Time to peak force as a percent of stance time among distinct shoeing conditions.

Time to (A) braking (PFB), (B) vertical (PFV), or (C) propulsion (PFP) peak force from horses (n = 6) shod with egg-bar, heart-bar, open- heel, or wooden clog shoes and while unshod. Points in the graph represent individual trials, and the horizontal line in the middle of each set indicates the mean; the numeric value along with the standard error of the mean (in parentheses) are shown to the right of each data set. Significant differences between data sets with a line between them are indicated by a symbol beneath the line.

Figure 7 Braking time as a percent of total stance time among shoeing conditions.

Percent (mean ± SEM) of stance time in braking from horses (n = 6) shod with egg-bar, heart-bar, open-heel, or wooden clog shoes and while unshod. Significant differences between data sets with a line between them are indicated by a symbol beneath the line. Mean and standard error of the mean (in parentheses) values are shown above each column.

Though not significantly different among shoeing conditions, the ST was shortest when unshod and longest with open-heel shoes (Fig. 8). The data point distribution was distinct among shoeing conditions. Most of the individual ST data points with open-heel and wooden clog shoes or while unshod were condensed around the medians, while those with egg-bar and heart-bar shoes were widely distributed above and below the median (i.e., higher interquartile range).

Figure 8 Stance time among shoeing conditions.

Box plots showing the stance time from horses (n = 6) shod with egg-bar, heart-bar, open-heel, or wooden clog shoes and while unshod. The values below each data set are the interquartile range (IQR), minimum value (Min), maximum value (Max), mean (Mean) and standard error of the mean (SEM).

Y-Z resultant force vectors

The resultant force vectors, GRFYZ, were initially directed caudally and transitioned to a cranial direction at about midway through the step cycle for all shoeing conditions, though the point of transition from braking to propulsion occurred a bit later in the step cycle (∼55%) with the wooden clog shoe compared to other shoeing conditions (∼52%) (Fig. 9). The vector magnitude had two peaks for all shoeing conditions, one at about 35% of the step cycle and another at 65% of the step cycle for the wooden clog shoe and at 60% of the step cycle for the rest. The magnitudes of the resultant vectors with the wooden clog shoe were less than the other shoeing conditions during the loading phase, greater at the first peak, less at the second peak, and greater during unloading. Also unique to the wooden clog shoe was a decrease in magnitude between the two peaks in contrast to the other shoes for which the magnitude plateaued slightly around the first peak but continued to increase throughout the distance between peaks. While unshod, the magnitude of GRFYZ was lower than the other shoeing conditions during the first peak. Significant differences included that the GRFYZ was less with the wooden clog versus egg-bar and hear-bar shoes at 5% of the step cycle and less than all other shoeing conditions at 10% and 15% of the step cycle (Table 1). It was greater with the wooden clog shoe compared to unshod at 30% and 35% of the step cycle. The magnitude was less with the wooden clog versus open-heel shoe at 50% of the step cycle, less with the wooden clog shoe versus all other shoeing conditions at 55%, and less than with egg-bar shoes and while unshod at 60%. It was greater with the wooden clog shoe than while unshod or with heart-bar or open-heel shoes at 70% of the step-cycle, and greater than all other shoeing conditions at 75–90% of the step cycle.

Figure 9 Y–Z resultant ground reaction force vector diagrams and magnitude.

Force vectors (upper panel) and vector magnitude (mean ± SEM, lower panel) from forelimbs of horses (n = 6) walking over a force platform embedded in a 40-m concrete runway in 5% increments of the complete step cycle while shod with egg-bar (A, red), heart-bar (B, brown), open-heel (C, lavender), or wooden clog shoes (D, green) or while unshod (E, black). In the upper panel each line is the mean value of all horses for the indicated increment and shoeing condition. Specific ranges of the step cycle are shown in shades of grey (0–15%), green (20–35%), orange (40–55%), blue (60–75%), and purple (80–100%), respectively.

Table 1 Fyz vector magnitude (mean ± SEM) for 5% step cycle increments.

GRFyz (N/kg)	Egg bar	Heart bar	Open heel	Wooden clog	Unshod	
Step cycle increment	Mean	SEM	Mean	SEM	Mean	SEM	Mean	SEM	Mean	SEM	
0	0.02	0.00	0.03	0.02	0.02	0.01	0.05	0.01	0.02	0.00	
5	1.34a	0.09	1.32a	0.04	1.24ab	0.09	0.96b	0.12	1.28ab	0.05	
10	2.57a	0.11	2.60a	0.14	2.55a	0.08	1.99b	0.10	2.62a	0.09	
15	3.70a	0.08	3.74a	0.03	3.72a	0.04	3.25b	0.11	3.64a	0.11	
20	4.45	0.04	4.47	0.01	4.47	0.06	4.16	0.15	4.34	0.07	
25	4.88	0.02	4.88	0.05	4.90	0.03	4.87	0.11	4.72	0.05	
30	5.12ab	0.06	5.13ab	0.08	5.15ab	0.02	5.29a	0.10	4.94b	0.00	
35	5.17ab	0.07	5.16ab	0.06	5.22ab	0.03	5.38a	0.10	4.96b	0.04	
40	5.19	0.06	5.16	0.01	5.27	0.06	5.29	0.10	5.00	0.03	
45	5.30	0.02	5.28	0.04	5.41	0.08	5.22	0.10	5.20	0.02	
50	5.60ab	0.03	5.59ab	0.07	5.68a	0.11	5.29b	0.09	5.59ab	0.06	
55	6.05a	0.07	6.02a	0.09	6.04a	0.14	5.58b	0.07	6.06a	0.08	
60	6.29a	0.08	6.21ab	0.06	6.16ab	0.17	5.91b	0.06	6.26a	0.04	
65	6.10	0.04	6.00	0.01	5.90	0.19	6.04	0.10	6.08	0.04	
70	5.56ab	0.02	5.47a	0.06	5.36a	0.19	5.86b	0.13	5.55a	0.12	
75	4.78a	0.08	4.75a	0.04	4.68a	0.14	5.45b	0.16	4.88a	0.17	
80	3.89a	0.12	3.94a	0.01	3.87a	0.05	4.79b	0.17	3.91a	0.13	
85	2.98a	0.05	2.96a	0.03	2.96a	0.02	3.90b	0.15	2.89a	0.03	
90	2.13a	0.03	2.16a	0.03	2.11a	0.05	2.73b	0.09	2.06a	0.00	
95	1.36	0.03	1.37	0.00	1.35	0.03	1.54	0.06	1.32	0.02	
100	0.03	0.01	0.03	0.00	0.03	0.00	0.04	0.00	0.03	0.00	
Notes.

Values with distinct superscripts within a row are significantly different among shoeing conditions (p-value < 0.05).

Shoeing condition classification with a random forest model

The PFB was highly correlated with IMPB (0.9, p-value < 0.001) as well as the PFB/PFV ratio (0.94, p-value < 0.001) (Fig. S2). The IMPB was highly correlated with the PFB/PFV ratio (0.89, p-value < 0.001). Similarly, PFP was highly correlated with IMPP (0.75, p-value < 0.001) and moderately correlated with percent ST to PFV (0.62, p-value < 0.001). The percent of ST to PFB and to PFP were moderately correlated (0.47, p-value < 0.001). The observed correlations were expected due to interdependence among variables.

The RF model could predict shoeing condition from kinetic measures with an accuracy of about 41%. The wooden clog shoe (F1-score = 1.0) could be predicted with higher positivity than the unshod and egg-bar shoes (F1-score = 0.38) which could be predicted with a higher positivity than heart-bar and open-heel shoes (F1-score ≤ 0.29) (Fig. 10A). The model had an asymptotic training error curve along the x axis at a tree depth higher than 7, and the classification error was between 0.8 and 0.6 up to that depth (Fig. 10B). Shoeing condition classification relied on all features included in the model but most heavily on percent ST to PFP,percent ST to PFV, and PFP (Fig. 10C). Taken together, these results indicate that wooden clog shoes were highly and accurately predictable with the RF model using the kinetic measures included in this study as features.

Figure 10 Random forest model outcomes.

(A) Confusion matrix for the RF model to predict shoeing condition from kinetic variables with each cell across the matrix diagonal showing the percent of true predictions and all other cells showing false predictions. (B) Training (blue) and test (red) error curves against different maximum tree depths in the RF model used to predict shoeing condition. (C) Feature importance (mean ± SEM) of each variable in the random forest algorithm classification model to predict shoeing condition from kinetic variables.

Discussion

Results of this in vivo study establish the effects of shoe configuration on equine gait kinetics at a walk. The first hypothesis, vertical, braking, and propulsion PF and IMP are different while shod with heart-bar, egg-bar, open-heel, and wooden clog shoes, or while unshod, was rejected since pairwise comparisons between shoeing conditions confirmed that not all measures were significantly unique. The second hypothesis, the resultant GRFYZ vector has the longest duration of cranial angulation with open-heel shoes followed by unshod, then egg-bar and heart-bar shoes, and the shortest with wooden clog shoes, was rejected because it was not supported by the GRFYZ vector angle distribution. In fact the cranial angulation was slightly longer with the wooden clog shoes. Although the hypotheses were rejected, the kinetic values compose a unique set of measures for each shoe configuration. Several force measures and most temporal parameters with wooden clog shoes were different from unshod and with the other shoes. Resultant GRFYZ vectors had a slightly broader distribution over the step cycle and a later transition from braking to propulsion with wooden clog shoes relative to other shoeing conditions. Variations in vector magnitude at distinct points during the step-cycle confirmed unique patterns among shoeing conditions with the highest number of significant differences between the wooden clog shoe and the others. Kinetic measures (features) had distinct, clear correlations with the target variable (shoeing condition) when horses were shod with wooden clog shoes compared to other shoeing conditions as indicated by the prediction accuracy of the RF classification. Results of this investigation provide a novel perspective about how equine gait kinetics at a walk are altered by application of the shoes in this study. This new information is useful to understand the kinetic impacts of shoe design.

The unique configuration and composition of the wooden clog shoe likely contributed to lower PFV given that there were no significant differences in stance time and speed to which to attribute the distinction. This could be a result of both the lower stiffness of the wood composition compared to the iron-based shoes and higher contact surface area (Fajdiga, Šubic & Kovačič, 2021; Moyer & Anderson, 1975). Reports assessing the direct effect of shoe stiffness on impact force and GRF magnitude in humans confirm that both increase with increasing shoe stiffness; the higher GRF magnitude impacts proximal joint moments and range of motion (Boyer et al., 2012; Lin et al., 2017; Teoh et al., 2013). A lower PFV with wooden clog shoes also likely contributed to the slightly higher PFB/PFV ratio, a representation of interactions between the hoof and ground surfaces, compared to unshod. Reducing the ratio indicates decreased grip, which could also be a benefit from less jarring during hoof deceleration (Parkes & Witte, 2015). The ratio values in this study are aligned with those in a previously published kinetic study, though ground and hoof surface properties contribute to differences among investigations (Chateau et al., 2010; Lewis et al., 2015; Pardoe et al., 2001; Parkes & Witte, 2015; Wang et al., 2021).

The efficacy of the beveled shape of the wooden clog shoe to facilitate breakover is supported by the highest percent stance in braking and lowest in propulsion in this study (O’Grady, 2020). Specifically, the beveled shoe shape is thought to shorten the breakover distance by moving it in a palmar/plantar direction like a rolled toe shoe (Van Heel, VanWeeren & Back, 2006). In a previous study, the distal interphalangeal joint moment arm during breakover was reduced by natural balance (rolled toe) and quarter-clip shoes compared to toe-clip shoes, but the peak distal interphalangeal joint moment was not significantly different between shoes (Eliashar et al., 2002). Additional research is necessary to investigate if the same is true with wooden clog shoes. The longer times to reach the peak of all forces with wooden clog shoes was also likely a consequence of the unique shape, which was unfamiliar to the horses in this study and has been shown to decrease the limb loading rate in humans wearing rocker-soled shoes compared to barefoot (Lin et al., 2017). Even though total stance time was not different, the rocking motion, and, potentially, the increased shoe height, required additional time to reach the peak force levels in all planes evaluated. Together, the study results confirm that the wooden clog shoe facilitates breakover and reduces peak vertical forces experienced by the limb during the second half of stance, effects that could benefit horses with distal joint pain and/or weakened hooves without greater force distribution to the rear limbs.

In contrast to the kinetic changes observed with the wooden clog, PF and IMP were fairly consistent among iron shoe shapes in this study as previously reported, though the results of this study are specific to forelimbs at a walk and results vary due to different gaits, velocities, and shoes among investigations (Amitrano, Gutierrez-Nibeyro & Schaeffer, 2016; Khumsap et al., 2002; Weishaupt et al., 2010). Changes in braking are especially meaningful given the greater braking function of the equine forelimbs (Dutto et al., 2004; Hobbs & Clayton, 2013). The higher IMPB with open-heel shoes versus unshod might be explained by the slightly higher PFB and percent braking time since the measure represents the area under the force versus time curve. The mass of the open-heel shoe may also have contributed to differences between the two conditions since a greater mass from the shoe can increase landing velocity of the hoof and result in a greater force at impact during the braking phase of the stance (Parkes & Witte, 2015).

The longer stance time with open-heel shoes could have contributed to the higher IMPV, despite a lower PFV. Though trial speed was not significantly different among shoeing conditions, it was slightly lower, 0.09 ms−1, with open-heel shoes compared to unshod, and it might have contributed to the higher IMPV and lower PFV values with the shoes. Another explanation is that the PFV point in this study occurred during the second half of the stance phase. The GRFYZ had a lower magnitude during the first peak while unshod compared to with open-heel shoes while the relationship reversed during the higher second peak. As such, and as indicated above, the PFV and IMPV values and comparisons reported here are specific to the second half of the stance. These points highlight the complexity of equine gait kinetics at a walk where the vertical forces have two peaks during the step cycle.

Graphs of the mean resultant vectors in the sagittal plane during the stance phase provide a broad perspective of variations in gait kinetics attributable to shoe configuration. The GRFYZ vector direction changes from caudal to cranial angulation at the point of full stance with unshod hooves, corresponding to horizontal decelerative followed by accelerative forces during stance (Hobbs & Clayton, 2013). The vector origin is affected by pressure distribution over the hoof surface, and it is impacted by alterations in pressure distribution over the hoof surface with shoes (Barrey, 1990; Parks, 2003); the vector direction depends on the magnitude of braking and propulsive forces. The results confirm unique shapes of the vector magnitude over the course of the step cycle and support the individual force data. Specifically, the magnitude was greatest with the wooden clog shoe during the first peak, but it was significantly lower during the higher second peak that corresponded to PFV. The distinct decrease in the magnitude between the two peaks with the wooden clog shoe is consistent with GRF forces applied across the larger, convex surface area and rocking motion during the transition from braking to propulsion. Additionally, the higher magnitudes with the wooden clog shoe during the propulsion phase are consistent with the higher PFP as is the lower magnitude during braking while unshod which had the lowest PFB and IMPB. A slightly delayed transition to propulsion with the wooden clog shoe coincides with the greater braking ST percentage with the shoe. Despite higher forces during the first half of the stance and higher propulsive forces during the second half, the lower PFV with the wooden clog shoe could reduce hoof stresses and benefit weakened or damaged tissues. Similarly, the smoother breakover could reduce distal interphalangeal joint extension, reduce deep digital flexor tendon strain, and decrease pressure on the navicular joint during the final part of the stance (Hagen et al., 2021). The sagittal plane resultant GRFYZ vector analysis in this investigation expands current understanding of the impact of shoes on kinetic forces experienced by the hoof and forelimb at a walk. Potential long-term implications of the vector distributions for the hoof and distal limb will require additional study.

Use of a RF machine learning algorithm to predict shoeing from kinetic measures determined in this study showed that the wooden clog shoe was classified with the highest accuracy. It is not surprising that the percent ST to PFV and PFP and the PFP had the highest importance for the model since the parameters tended to be among the most distinctive for the wooden clog shoe. The model did not perform well for the other shoeing conditions as they all had poor F-1 scores (less than 0.5). The performance could be related to an insufficient number of data points in the training phase of the model, especially since the measures were relatively similar among shoeing conditions apart from the wooden clog shoe. Similar values within features for different target variables can negatively impact the accuracy of machine learning algorithms, so increasing the number of features and feature data points might improve model accuracy. Highly correlated features can decrease model accuracy; removal of some features might improve model performance. The fact that shoeing conditions could be distinguished from each other using the data from this single study indicates that machine learning algorithms could provide another data analysis mechanism that could substantially expand current knowledge of the relationship between shoe configuration and gait kinetics.

This study was limited by the number of horses included, six light breed horses between 5 and 25 years old. Though the horses were objectively and subjectively not lame, variations in kinetic measures due to subtle limb or hoof pain cannot be entirely ruled out. This could have contributed to intra- or inter-individual differences that reduced statistical power. The study results might also not apply to all horses due to natural variability, including gait variations, among biological subjects. The speed range in this study was relatively wide compared to other studies, though speed did not vary significantly among trials. Hence, the range includes natural variation in a comfortable walking speed among horses included in the study. The results are limited to forelimbs and the shoes included in the study at a walking gait, and peak kinetic values to the second half of the stance. Shoes with similar configurations but with different thickness, mass, or material composition could influence the gait differently (Pardoe et al., 2001; Rogers & Back, 2003; Wang et al., 2021). The study results also do not take into consideration the breathability of the tested horseshoes, and this property is critical in the design of prosthetics and shoes (Aflatoony, 2019; Pereira et al., 2007). Effects of shoe material and shape on shear forces, hoof stability, and joint torque, especially given the distinct height and composition of the wooden clog shoe, are important outcomes to be considered for future study designs. A longer period of time with each shoe configuration, typical of a standard shoeing interval, might have allowed for more acclimation and could have resulted in fewer differences among shoes (Heel, Weeren & Back, 2006). Additionally, the ground surface for gait trials in this study was concrete. Different ground surfaces will likely have different results (Crevier-Denoix et al., 2010; Robin et al., 2009).

Conclusion

The results of this study provide novel information and unique demonstrations of the impact of shoe shape on equine gait kinetics. The distinctive and highly predictable changes in gait kinetics with the wooden clog shoe were clear, supporting their use to reduce forces experienced by hooves and distal limbs during the second half of stance and during breakover, though higher forces during the first half of stance and during propulsion are important to consider. The altered gait kinetics could potentially protect laminitic hooves and alleviate discomfort from distal interphalangeal joint, distal deep digital flexor tendon, and/or navicular bone pathology. Additionally, machine learning algorithms might eventually improve the ability to design and select shoes intended for hoof protection and function by confirming that they have sufficient impact on target gait kinetics to be identified. Taken together, the results of this study enhance knowledge of shoe effects on equine kinematics at a walk and highlight cutting-edge tools to quantify them.

Supplemental Information

Supplemental Information 1 Trial speed across shoeing conditions

Box plots illustrating speed of gait trials with horses (n = 6) shod with egg-bar, heart-bar, open heel, or wooden clog shoes and while unshod. The values below each data set are the interquartile range (IQR), minimum value (Min), maximum value (Max), mean (Mean) and standard error of the mean (SEM).

Supplemental Information 2 Study variables correlation matrix

Correlation coefficients between equine forelimb kinetic gait variables recorded from non-lame horses (n = 6) at a walk. The scale on the right side of the matrix shows the colors on the graph that represent the correlation coefficients among the variables ranging from −1 (dark blue), through 0 (light blue), to +1 (off-white). Asterisks above each correlation coefficient show the Pearson’s test for correlation p-value (∗p-value < 0.05, ∗∗p-value < 0.01, ∗∗∗p-value < 0.001).

Supplemental Information 3 Raw kinetic data from individual gait trials

Each line in the file contains data from individual trials for each horse and shoeing condition used for force and vector calculations. Measures were always normalized to weight then averaged based on shoe types. Shoeing conditions were used as a fixed effect for all statistical comparisons in mixed models.

This work was only possible with the help of previous and current LECOR members with data collection. The authors also acknowledge Ms. Cindy Meeker for her professional farrier work, Drs. David Schaeffer and Chin-Chi Liu for assistance with statistical analysis, and Mr. Michael Keowen and Mr. John-Ross Miller for their help with horse husbandry.

Abbreviations

PFV Peak vertical force (N kg−1)

PFB Peak braking force (N kg−1 )

PFP Peak propulsion force (N kg−1)

IMPV Vertical impulse (N s kg−1)

IMPP Propulsion impulse (N s kg−1)

PFB/PFV ratio Ratio of braking to vertical peak forces (–)

ST Stance Time (s)

Additional Information and Declarations

Competing Interests

Author Contributions

Animal Ethics

Data Availability

The authors declare there are no competing interests.

Rita Aoun conceived and designed the experiments, performed the experiments, analyzed the data, prepared figures and/or tables, authored or reviewed drafts of the article, and approved the final draft.

Zaneta Ogunmola performed the experiments, analyzed the data, prepared figures and/or tables, and approved the final draft.

Anaïs Musso performed the experiments, analyzed the data, authored or reviewed drafts of the article, and approved the final draft.

Takashi Taguchi performed the experiments, analyzed the data, prepared figures and/or tables, and approved the final draft.

Catherine Takawira performed the experiments, authored or reviewed drafts of the article, and approved the final draft.

Mandi J. Lopez conceived and designed the experiments, performed the experiments, analyzed the data, prepared figures and/or tables, authored or reviewed drafts of the article, and approved the final draft.

The following information was supplied relating to ethical approvals (i.e., approving body and any reference numbers):

The Louisiana State University Institutional Animal Care and Use Committee provided careful review and approval for this research (Protocol: IACUCAM-21-157). Based on Animal Welfare Assurance #A3612-01, License #72-3; Multiple Assurance #M1128.

The following information was supplied regarding data availability:

The raw data are available in the Supplemental Files.

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
