# Peer review of "Shoe configuration effects on equine forelimb gait kinetics at a walk"

_PeerJ, doi:10.7717/peerj.18940_

## Round 0.1 · original submission · Minor Revisions

Please address the comments by the reviewers. I apologise for the time it has taken to get a sufficient number of reviews on this occasion.

The deadline given by PeerJ is 10 days - however I think the revisions may warrant more time, so please do not worry if the revisions take longer.

·

Basic reporting

No concerns. Well written and nicely strucutred with ample of references.

Experimental design

No major concerns. Please see 'additional comments' for specific comments/suggestions about the materials and methds section.

Validity of the findings

Thank you to the authors for reanalzying the data. Very nice read now.

There is one additional thing that would make this manuscript even more useful. I have commented on this withiin the 'additional comments' in more detail. When lookign at the revised results in particular figure 9 with the yz forces, it looks like that while peak vertical forces are reduced in the second half of stance, they may actually increase in the first half of stance (the clog cannot reduce the vertical force as such). This would be super interesting to analyze. If this is not possible, then at the very least, it needs to be made clear, that the current study investigates the peak vertical forces in the secnd half of stance, which is critical to indicate in the context of 'walk' where typically two peaks are observed during the stance phase.

Additional comments

Here a list of comments as well as a general remark. Thank you for revising this manuscript!

General: A huge thank you to the authors taking on board my comments of a previous round of reviews and reanalyzing the data. The data make so much more sense now. I have a few additional comments, mainly suggestions in relation to data interpretation with most relevance to the discussion section and a couple of comments relevant for the materials and methods.

Detailed comments:
Materials and methods:
1) Would it be possible to make it more obvious in the materials and methods section that the shoeing interventions had been applied to the front limbs? And clarify what exactly had been done to the hind limbs. The latter might be for example relevant in the context of the front/hind distribution.
2) When attaching the metal shoes as indicated these were attached with two nails on either side. Were the shoes ‘clenched up’ at all at this stage. One would assume so given that the measurements were done 24h later. For completeness of the description of the methods, this additional piece of information would be helpful to add to the materials and methods.

Discussion:
Line 371 to 386: It would be super helpful if the reader would not have to go back to the introduction to check out the hypotheses. Please make it clear how exactly the results provide support for the hypotheses or how they present information that is opposed to the hypotheses.
Line 381-383: “Resultant force vectors … had … lower magnitude along the z axis …”. Looking at figure 9, which presents the YZ forces in a very nice manner, it almost looks like, that while the z-forces are lower in the second half of stance with the wooden clog, they are actually higher in the first half of stance in this shoeing condition. A couple of comments here:
Would it be possible to test whether there is a difference between the z-force peaks in the first half of stance as well?
This also relates to the following paragraph (line 387-399) which is discussing topics around the assumption that the vertical force is lowered. The reduction of vertical forces is a topic often discussed by farriers and a more complex topic than might appear at first sight. As such it would seem important to present the ‘entire picture’ here. The vertical force production is governed by gravity requiring the animal to provide body support. As such there is given amount of vertical force that needs to be produced. Vertical force cannot ‘disappear’. So the very interesting question here is: how do the horses achieve this reduction in peak force. The authors mention stance time and speed as important factors. The ‘shape’ of the ground reaction force curve is an additional parameter that can influence the peak force when speed and stance time are unchanged. In this case it might look like that exactly this is the case: the horses produce higher peak vertical forces in the first half of stance and lower peak vertical forces in the second half of stance with the clog. This is super interesting (if indeed confirmed by statistics).
This might also go some way into explaining figure 5 and the lower front/hind ratio for the wooden clog if this ratio id based on the peak values identified in the second half of stance. What do these ratios look like for the first half of stance?
Line 413-416: “Together, the study results confirm that the wooden clog shoe facilitates breakover and reduces peak vertical forces experienced by the limb, effects that could benefit horses with acute distal joint pain and/or weakened hooves without greater force distribution to the rear limbs.” This sentence could do with some clarification in relation to the ‘reduction of peak vertical forces’ (achieved in the second half of stance!) as well as the ‘distribution to the rea limbs (again this only holds true for the second half of stance based on the current data).
Line 427-428: how does the mass of the shoe play a role when the limb is in stance and hence in contact with the ground?
Conclusion 484-486: “The distinctive and highly predictable changes in gait kinetics with the wooden clog shoe were clear, supporting their use to reduce forces experienced by hooves and distal limbs.” Please rephrase to be more specific, for example “reduce peak vertical force in the second half of stance.”

·

Basic reporting

A useful study comparing the effect of different shoe types on kinetics and temporal variables during stance. The manuscript is generally well written, although better justification for performing this work is needed initially.
Line 83-87: It is not completely clear that these shoes are designed to modify traction characteristics. Please make this clearer.
Line 95-99: Please justify your hypotheses from the previous studies you used to develop them. Hypothesis 1 is also rather general.
Line 116: I am interested to know what your reasoning was for selecting these specific shoes. It would be useful to expand your introduction to include a reasoning/justification for selecting these specific shoe configurations.

Experimental design

In my opinion, I do not think the modelling is appropriate for such a small sample size.
Line 168: What was your acceptable variance in individual horse walking speed between conditions?
Line 238-243: As you have only used 6 horses in this study, but used multiple trials from each horse/condition to increase your sample size I am not convinced that the Random Forest analysis is appropriate. It is not clear why you might try to predict the outcome of a shoeing condition from the kinetic data particularly, as you only have a small number of different shoe types with wooden clogs being completely different to the other conditions. Also, which trials did you use for training compared to ones you used for testing?

Validity of the findings

The discussion could be improved with greater focus on how the results might be of benefit for clinical applications (which links back to the justification for the study in the first place). Please see detailed comments below.
Line 246: The open heel shoes are quite noticeably slower than the other conditions. Can you explain why.
Line 323-328: Where are your statistical results for resultant vector length and angle for the whole stance phase? I can see you have included vector diagrams, which are useful, but the metrics from these diagrams do not appear to have been analysed statistically.
Line 353-354 and 363-364: Contradiction here? I can’t see really how a model can be inaccurate and also highly accurate.
Line 372 -375: This comes back to the fact that hypothesis 1 is too general. You are saying that there are some differences here, but again this is a generalization.
Line 376: Please test the hypothesis correctly using statistics and not observation.
Line 385: by this shoe application.
Line 394-396: This statement does not make sense or correspond with your data? Also in this paragraph, what is the benefit of reducing the braking forces/frictional characteristics? Under what circumstances/application?
Line 405: I don’t follow your reasoning here. If the DIPJ moment is not different what are the unintended consequences of reducing the moment arm?
Line 410-419: The extra length of the limb may be influential here. Can you consider how this may be relevant to a change in timing of force peaks? It may also be useful to look at human studies using rocker types of shoes.
Line 426-427: The vertical impulse differences should not be explained in this way. Your peak vertical forces are lower in open heeled and stance time is longer. As there is a difference in speed it concerns me that this is just a speed effect. How do you explain a smaller peak force compared to unshod otherwise?
Line 437-444: As you mentioned that wooden shoes could be beneficial for horses with joint issues, how could the overall changes seen in the discrete variables and force vector patterns help horses with such issues based on what we know about the timing of joint motions through the stance phase.
Line 455-456: From the data you have presented the spread is either the same (which I don’t really see) or greater in the plots for the wooden clogs. If you want to discuss within horse consistency, you need to present the data somewhere. There seems to be quite large between horse differences in the wooden clogs. The RF model does not show these data adequately.
How does your discussion build upon the work of Aoun et al., 2023?
Figures 3 and 4: Your SD’s in these diagrams do not appear to match the spread of data or the discussion of them?

Reviewer 3 ·

Basic reporting

Introduction
The introduction section is a little bit extensive, for example: It does not seem to be necessary to describe exactly the configuration of every shoe (heart-bar; egg-bar and wooden), once it is well described in the literature. This would allow a more fluently reading. Additionally, the measures pointed to describe the three models of shoes can vary significantly among different situations, diseases or animal size. It is suggested to emphasize information about the forces acting on the hoof.
The literature references are substantial, although some of them are outdated.
The structure, figures, tables and background are ok.

Experimental design

Was a sample size testing performed? If yes, I believe that it is important for make the results more consistent.
It was used a 25 years old horse; it is very difficult to find a horse old like that presenting no lameness at all.
How was the hooves condition before the basic trim? How many days before the study the horses were shoed? Even when done properly, shoeing can cause a locomotion alteration in terms of kinetic and kinematics of the horse.
The hypothesis is well defined and explained, and the statistical method as well. The study is well described so it could be replicated.

Validity of the findings

Since all the shoes presented in the study are widely applied to many situations at the equine medicine, the impact of the information is enormous. The results can help and veterinarians all around the world to define which shoe is better for different situations.
Conclusion is direct and objective.

Additional comments

The study is very interesting and elucidates many questions that equine practitioners may have on the day to day routine. It is not easy to perform in vivo experiments that uses vetorial forces.
The paper is substantial for sure.

---

## Round 0.2 · accepted · Accept

Thank you for the extensive revisions to this manuscript, it can now be accepted for publication. While this has taken some time, I think that it's a good example of how the review system should work, as I feel the review process has helped improve the paper and I'm aware that it allowed some adjustments to an analysis protocol. Thank you for your patience with the process.

·

Basic reporting

No comment

Experimental design

No comment

Validity of the findings

No comment

Additional comments

The authors have made substantial changes based on the review comments. This manuscript, in my opinion, is now much improved and a valuable resource for practitioners. I congratulate the authors on their efforts.